# LIGHTWEIGHT UNCERTAINTY MODELLING USING FUNCTION SPACE PARTICLE OPTIMIZATION

## ABSTRACT

Deep ensembles have shown remarkable empirical success in quantifying uncertainty, albeit at considerable computational cost and memory footprint. Meanwhile, deterministic single-network uncertainty methods have proven as computationally effective alternatives, providing uncertainty estimates based on distributions of latent representations. While those methods are successful at out-of-domain detection, they exhibit poor calibration under distribution shifts. In this work, we propose a method that provides calibrated uncertainty by utilizing particle-based variational inference in function space. Rather than using full deep ensembles to represent particles in function space, we propose a single multi-headed neural network that is regularized to preserve bi-Lipschitz conditions. Sharing a joint latent representation enables a reduction in computational requirements, while prediction diversity is maintained by the multiple heads. We achieve competitive results in disentangling aleatoric and epistemic uncertainty for active learning, detecting out-of-domain data, and providing calibrated uncertainty estimates under distribution shifts while significantly reducing compute and memory requirements[1].

## 1 INTRODUCTION

Deep learning is becoming ubiquitous in our lives, with applications ranging from medical diagnosis to autonomous driving. However, in safety-critical scenarios accurate predictions alone are not sufficient. In addition, models should provide well-calibrated uncertainty estimates to mitigate overconfidence and the potential risks associated with erroneous predictions. Uncertainty methods in deep learning typically distinguish between two types (Kendall & Gal, 2017; Hüllermeier & Waegeman, 2021): (a) Aleatoric uncertainty, which arises from inherent data ambiguity or noise, and (b) epistemic uncertainty, which corresponds to model uncertainty resulting from a lack of knowledge and observations.

Much research is concerned with reliably estimating epistemic uncertainty in neural networks. Bayesian neural networks (BNNs) offer a theoretically sound solution by treating model parameters as probability distributions rather than point estimates, providing a fully stochastic model (Neal, 1995; MacKay, 1992). However, accurately approximating the posterior distribution remains challenging (Izmailov et al., 2021). It often requires multiple forward passes and significant computational overhead (Blundell et al., 2015; Gal & Ghahramani, 2016; Dusenberry et al., 2020).

In contrast to BNNs, deep ensembles represent a conceptually simple empirical method that has often outperformed Bayesian methods in practical scenarios (Lakshminarayanan et al., 2017; Ovadia et al., 2019). The prevailing method for training deep ensembles is to randomly initialize the network parameters and to perform independent optimization of the ensemble members. During inference, the ensemble members' predictions are combined and uncertainty estimates are derived from the disagreement between these predictions. However, there is no guarantee for functional diversity among the ensemble members (Abe et al., 2022b;a).

Recent research in particle-based variational inference addresses this lack of diversity in deep ensembles and offers a principled way of viewing deep ensembles from a Bayesian perspective (Liu & Wang, 2016; Liu, 2017; Liu et al., 2019; D'Angelo & Fortuin, 2021). These methods approximate

---

[1]Code will be made publicly available upon acceptance.

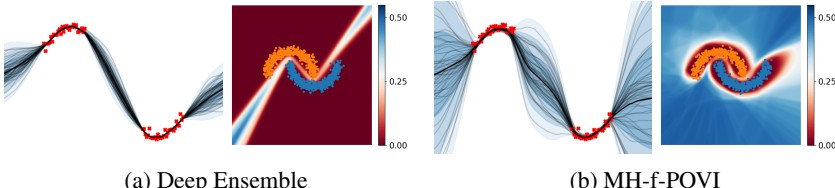

(a) Deep Ensemble          (b) MH-f-POVI

Figure 1: Predictions of deep ensembles and the proposed multi-head (MH) network with function space loss (MH-f-POVI). For regression, we show the prediction of single particles, the mean and the standard deviation. For classification on the two-moons data, we show the standard deviation of the predicted probabilities $p(\mathbf{y} \mid \mathbf{x}, \theta)$. Deep ensembles are overly confident in regions without training data, while MH-f-POVI predictions are enforced to be diverse outside of the training data.

the posterior distribution of the neural network parameters using a set of discrete particles. For deep ensembles, this entails modifying the optimization procedure by incorporating a kernelized repulsion term to prevent ensemble members from converging to the same optimum (Wang et al., 2019; D'Angelo & Fortuin, 2021). However, due to the over-parameterized nature of neural networks, particles can converge to solutions that are distant in the parameter space but the networks still represent the same function. Wang et al. (2019) avoid this issue by performing particle inference directly in the space of functions, explicitly enforcing functional diversity of ensemble members. Nevertheless, a remaining and significant limitation of deep ensembles lies in the computational cost and memory requirements during inference, that practically limits the number of particles.

In this paper, we address this by utilizing the fact that particle inference in function space does not impose any restrictions on the choice of the parameterization model, other than sufficient flexibility. Inspired by ensemble distillation methods (Tran et al., 2020), we propose a multi-headed network architecture, where each head represents a particle in function space, illustrated in Fig. 2. Thus, we can still leverage the theoretical framework of particle-based inference while avoiding the need for maintaining the full deep ensemble. By utilizing a base network that learns a shared representation, we are highly effective in terms of computational and memory cost. At the same time, the ensemble heads are enforced to provide diverse predictions through the function-space repulsion loss.

In summary, our contributions are as follows:

- We propose a simple and scalable uncertainty estimation method based on function-space particle inference. Instead of using full deep ensembles, we advocate a single network with multiple heads to be significantly more parameter-effective.

- We demonstrate that the repulsive ensemble head can be used to provide reliable retrospective uncertainties for a pre-trained network, given that the latent space is informative and well regularized (Miyato et al., 2018; van Amersfoort et al., 2021).

- We empirically evaluate our method for regression and classification tasks on synthetic and real-world datasets. We show that our network is able to (i) disentangle aleatoric and epistemic uncertainty for active learning, (ii) improve detection of both near and far out-of-distribution data, (iii) provide calibrated uncertainty estimates under distribution shifts.

## 2   BACKGROUND

We consider supervised learning tasks. Let $\mathcal{D} = \{\mathbf{x}_i, \mathbf{y}_i\}_{i=1}^N = (\mathbf{X}, \mathbf{Y})$ denote the training data set consisting of $N$ i.i.d. data samples with inputs $\mathbf{x}_i \in \mathcal{X}$ and targets $\mathbf{y}_i \in \mathcal{Y}$. We define a likelihood model $p(\mathbf{y}|\mathbf{x}, \theta)$ with the mapping $f(\cdot; \theta) : \mathcal{X} \to \mathbb{R}^K$ parameterized by a neural network.

**Bayesian neural networks.** The core concept of Bayesian neural networks (BNNs) involves the treatment of network parameters $\theta$ as random variables instead of point estimates. This entails defining a prior distribution of the parameters $p(\theta)$ to infer the posterior distribution of the parameters $p(\theta|\mathbf{X}, \mathbf{Y}) \propto p(\theta)p(\mathbf{y}|\mathbf{x}, \theta)$. Predictions for a test data point $\mathbf{x}_{\text{test}}$ are obtained by marginalizing over all possible parameters: $p(\mathbf{y}_{\text{test}}|\mathbf{x}_{\text{test}}, \mathbf{X}, \mathbf{Y}) = \int_{\Theta} p(\mathbf{y}_{\text{test}}|\mathbf{x}_{\text{test}}, \theta)p(\theta|\mathbf{X}, \mathbf{Y})d\theta$. However, when con-

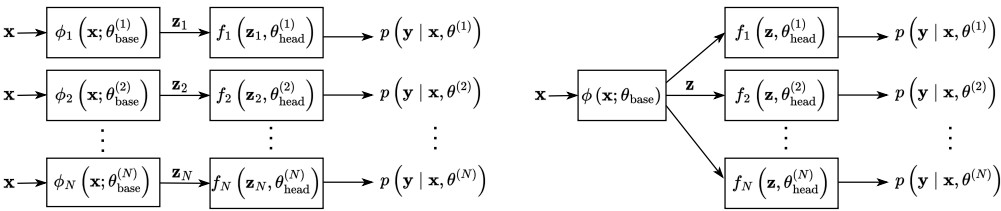

Figure 2: Deep ensembles *(left)* versus our MH-f-POVI multi-head architecture *(right)*. Diverse predictions are achieved by a repulsive deep ensemble on top of a shared base network.

sidering deep neural networks this integral is generally intractable. This sparked the development of various methods to approximate the posterior distribution, as summarized in Section 4.

**Particle optimization-based variational inference.** Variational inference aims to approximate the high-dimensional posterior with a simpler distribution. Particle optimization-based variational inference (POVI) methods (Liu & Wang, 2016) approximate the posterior distribution using a discrete set of particles $\{\theta^{(i)}\}_{i=1}^n$ in the form of $q(\theta) \approx \frac{1}{n} \sum_{i=1}^n \delta(\theta - \theta^{(i)})$, where $\delta(\cdot)$ is the Dirac function. Similar to traditional variational inference methods, the KL-divergence between the approximating distribution $q(\theta)$ and the true posterior $p(\theta|\mathbf{X}, \mathbf{Y})$ is minimized. However, particle methods offer greater flexibility due to their non-parametric nature and ability to cover multiple modes of the posterior. The particles can be efficiently optimized by following a deterministic gradient flow in a specified metric space that can be described by a partial differential equation (PDE) of the form $\partial_t q_t = -\nabla \cdot (\mathbf{v} \cdot q_t)$, where $\mathbf{v}$ represents the gradient flow. Utilizing a discrete set of particles simplifies this PDE to a discretized ordinary differential equation that can be solved iteratively

$$\theta_{l+1}^{(i)} \leftarrow \theta_l^{(i)} - \epsilon_l \mathbf{v}(\theta_l^{(i)})$$

where $\epsilon_l$ is the step size at time step $l$. A popular example of a POVI method is Stein variational gradient descent (SVGD) (Liu & Wang, 2016). In follow-up work, (Liu et al., 2019) provided further theoretical insights to particle optimization by viewing it as gradient flows in a Wasserstein space. Similarly, D'Angelo & Fortuin (2021) derived a gradient flow based in the Wasserstein space using kernel density estimation (KDE) of the gradient of the particle measure

$$\mathbf{v}(\theta_l^{(i)}) = \nabla_{\theta_l^{(i)}} \log \underbrace{p(\theta_l^{(i)} \mid \mathbf{x})}_{\text{POSTERIOR}} - \underbrace{\frac{\sum_{j=1}^n \nabla_{\theta_l^{(i)}} k\left(\theta_l^{(i)}, \theta_l^{(j)}\right)}{\sum_{j=1}^n k\left(\theta_l^{(i)}, \theta_l^{(j)}\right)}}_{\text{REPULSION TERM}}. \tag{1}$$

Instead of independent training, the particles are interacting with each other. In the limit of one particle, the training procedure reduces to standard MAP training, whereas in the infinite limit the approximation converges to the true posterior (Liu, 2017). Deep ensembles are a special case of particle methods where each particle is optimized independently without a repulsion term. In this paper, we use the KDE-WGD updating rule in Eq. (1) with a Laplacian repulsion kernel $k(x_1, x_2) = \exp\left(-\frac{\|x_1 - x_2\|}{\nu}\right)$ where $\nu$ is estimated using the median heuristic.

## 3 FUNCTION-SPACE PARTICLE OPTIMIZATION IN A SINGLE NETWORK

Applying POVI methods to neural networks does not necessarily result in the expected improvements. Due to the over-parameterization of neural networks, parameter configurations that are far away from each other can still yield the same functional behavior. To address this, Wang et al. (2019) moved posterior inference from parameter space directly to function space. In this case, the $n$ particles represent functions $f^{(1)}(\mathcal{X}), \ldots, f^{(n)}(\mathcal{X})$ that are updated according to

$$f_{l+1}^{(i)}(\mathcal{X}) \leftarrow f_l^{(i)}(\mathcal{X}) - \epsilon_l \mathbf{v}(f_l^{(i)})(\mathcal{X}). \tag{2}$$

Performing inference in the infinite-dimensional function space involves several approximations. Firstly, the function space is parameterized by any arbitrary flexible network $f(\mathcal{X}; \theta_l)$. Wang et al.

(2019) and D'Angelo & Fortuin (2021) kept the deep ensemble structure and parameterized each function with a separate neural network. Due to the interaction of the particles through the repulsion kernel, all ensemble members must be trained in parallel. This increases the computational demand for an increased number of particles.

**Multi-headed network structure.** Instead of using deep ensembles, we propose to use a shared base network with multiple heads that represent the particles in function space (see Fig. 2), i.e.

$$f^{(i)}(\mathbf{x}; \theta_{\text{base}}, \theta_{\text{head}}^{(i)}) = f_{\text{head}}^{(i)}(\phi(\mathbf{x}; \theta_{\text{base}}); \theta_{\text{head}}^{(i)}).$$

We call our proposed multi-headed network *MH-f-POVI*. Our model is highly parameter-effective by sharing the latent representation of the base model $\phi(\mathbf{x}; \theta_{\text{base}})$. Diverse predictions of the ensemble heads are enforced by the function-space repulsion loss. The base network $\phi(\mathbf{x}; \theta_{\text{base}})$ must be sufficiently powerful to learn meaningful representations of the underlying data. For regression tasks, the neural network heads parameterized by $\theta_{\text{head}}^{(i)}$ must be expressive enough to describe the predictive distribution and to produce diverse predictions in regions of low data density. For classification tasks, the ensemble heads represent a set of decision boundaries. To enforce diverse predictions in low-density regions of the feature space, the ensemble heads must be able to generate nonlinear decision boundaries.

**Retrospective uncertainties.** The multi-headed network approach provides a principled way of computing retrospective uncertainties given a *pre-trained* base network by separately training the base network and the repulsive ensemble heads. Discriminative models that lack proper regularization are prone to feature collapse, i.e., different data points that are distant in input space collapse into indistinguishable parts in feature space (van Amersfoort et al., 2020). To mitigate this, one effective approach is to promote distance-aware representations using bi-Lipschitz constraints: $K_L d_I(\mathbf{x}_1, \mathbf{x}_2) \leq d_F(\phi(\mathbf{x}_1), \phi(\mathbf{x}_2)) \leq K_U d_I(\mathbf{x}_1, \mathbf{x}_2)$. Here, $d_I$ and $d_F$ represent distance measures in the input and feature spaces, while $K_L$ and $K_U$ are the lower and upper Lipschitz constants, respectively. Miyato et al. (2018) introduce spectral normalization and residual connections as effective means to achieve these constraints. This avoids feature collapse while introducing smoothness (upper Lipschitz constant) in feature space.

**Choice of context points.** Performing the update rule in (2) involves evaluating the parameterized function over the entire set of $\mathcal{X}$. Wang et al. (2019) propose a mini-batch version that replaces the evaluation over the entire set $\mathcal{X}$ by drawing $B$ context points $\mathbf{x}_C$ from an arbitrary distribution supported on $\mathcal{X}^B$. In high-dimensional input spaces, it may not be practical to draw random samples from the input space. D'Angelo & Fortuin (2021) use the unmodified training data itself as context points, which may explain limited empirical results and underfitting of the function space methods. Rudner et al. (2022; 2023) use samples from datasets that are meaningfully related to the task, e.g. kMNIST for classification on MNIST and CIFAR100 for classification on CIFAR10. In addition, we artificially generate context points by negative data augmentation of the training data. For image classification tasks we create context points by randomly shuffling image patches.

## 4 RELATED WORK

**Efficient Bayesian neural networks.** A number of research has been invested in finding efficient methods to approximate the posterior distribution and to make BNNs scalable (Blundell et al., 2015; Gal & Ghahramani, 2016; Kingma et al., 2015; Dusenberry et al., 2020; Kristiadi et al., 2020). However, these approaches require multiple forward passes during inference, which can be limiting in time-critical tasks. Kristiadi et al. (2020) use Laplace approximation only for the weights of the last layer of a ReLU network and reduce overconfidence. Daxberger et al. (2021); Sharma et al. (2023) show that partially partially stochastic networks can achieve the same expressivity as the fully stochastic counterpart. Still, those networks require multiple forward passes. Our multi-headed structure can be interpreted as a partially stochastic network where the last layers are trained using particle optimization.

**Function-space inference.** A number of variational inference methods consider inference in function space. Sun et al. (2019) perform function-space variational inference in BNNs by taking the

supremum of marginal KL divergences over all finite sets of measurement points. Rudner et al. (2022) estimate the functional KL divergence by linearizing the function mapping of the neural network around a Gaussian distribution. Rudner et al. (2023) show that function-space regularization improves uncertainty estimates of neural networks both on in-distribution and out-of-distribution data. Wang et al. (2019) approximate the posterior distribution using a discrete set of particles that is optimized using Stein variational gradient descent. D'Angelo & Fortuin (2021) emphasize the connection between particle-based inference and deep ensembles. Both approaches parameterize the function space with deep ensembles, which imposes large computational and memory costs and limits the number of particles for practical use.

**Multi-headed architectures.** Various approaches have leveraged multi-headed network architectures to reduce memory footprint by sharing parameters of a base network (Song & Chai, 2018; Sercu et al., 2016; Lee et al., 2015). Osband et al. (2016) apply bootstrapping to a multi-headed network for exploration tasks in reinforcement learning. Zhu et al. (2018) perform online distillation of a teacher model using a multi-headed network. Tran et al. (2020) study the use of a multi-headed network to distill the functional diversity and uncertainty prediction of a given deep ensemble. Valdenegro-Toro (2023) ensemble a selection of layers for uncertainty prediction without a function space perspective.

**Deterministic single model uncertainty.** Single-model deterministic uncertainty methods (DUMs) have emerged as practical alternatives due to the computational costs and scaling limitations associated with BNNs and deep ensembles. The core idea of these methods is to consider the epistemic uncertainty of a given test input as being proportional to the distance to the support of the training data. If a test input is close to data points observed during training, the predictions are considered trustworthy. If the distance is large, the model should abstain from making predictions. Computing distances directly in high-dimensional input spaces, however, is often impractical. Thus, most methods depend on well-informed latent representations of the network and estimate epistemic uncertainty by considering feature space densities (Charpentier et al., 2020; Postels et al., 2020; Mukhoti et al., 2023; Winkens et al., 2020) or distances (Liu et al., 2020a; van Amersfoort et al., 2020). van Amersfoort et al. (2021) show that it is necessary to regularize the feature space appropriately to avoid feature collapse and to ensure that densities and distances are meaningful. Common methods to achieve bi-Lipschitz conditions include gradient penalties (Gulrajani et al., 2017; van Amersfoort et al., 2020), and spectral normalization (Miyato et al., 2018; Liu et al., 2020a). Prior networks (Malinin & Gales, 2018) model the predictive distribution of a model using a Dirichlet distribution, which involves a strong parametric assumption. Close to our approach are orthonormal certificates (Tagasovska & Lopez-Paz, 2019), that consist of a set of functions on top of a pre-trained feature extractor that map training samples to zero. Diversity is enforced by orthonormality conditions. Although these methods have shown strong performance in detecting out-of-domain data, calibrated uncertainty estimates under distributional shifts, such as corrupted data, are often neglected. Postels et al. (2021) demonstrate in extensive experiments that relying solely on the feature space density of a model is not necessarily indicative of the correctness of a prediction and results in poor calibration under distribution shifts.

## 5 EXPERIMENTS

We evaluate our proposed multi-headed architecture on several benchmark tasks. First, we start with an illustrative evaluation on a synthetic regression and classification problem. Then, we test its effectiveness in discriminating between aleatoric and epistemic uncertainty through experiments on MNIST including ambiguous images (Dirty MNIST) (Mukhoti et al., 2023). Furthermore, we perform out-of-distribution (OOD) detection on CIFAR10/CIFAR100 (Krizhevsky, 2009) for both far (SVHN (Netzer et al., 2011)) and near (TinyImagenet (Le & Yang, 2015)) OOD data. We use the uncertainty decomposition for active learning in Appendix A.2. Finally, we evaluate calibration of uncertainty estimates under various synthetic corruptions. See Appendix A.1 for a summary of the training details and further ablation studies.

**Synthetic Data** We illustrate the effectiveness of the multi-head structure as a lightweight parameterization and the benefits of performing inference in function space on two toy examples. We estimate the epistemic uncertainty for a one-dimensional regression and a two-dimensional classi-

fication problem using full deep ensembles and MH-f-POVI. As a base network, we use a feed-forward neural network with 3 hidden layers and 128 neurons. The repulsive head consists of 30 particles without hidden layers for regression and 1 hidden layer with 20 neurons for classification. Results are shown in Fig. 1. Deep ensemble uncertainty estimates are not representing the data density. By performing particle inference in function space, we are able to obtain reasonable uncertainty estimates with a much simpler network structure.

**Baselines.** We report uncertainties obtained by a single network with softmax output (*Single model*), and a deep ensemble consisting of 5 randomly initialized networks (*5-Ensemble*). Additionally, we evaluate Deep Deterministic Uncertainties (*DDU*) (Mukhoti et al., 2023) as a simple yet powerful representative for DUMs. DDU fits a classwise Gaussian using Gaussian Discriminant Analysis (GDA) to feature space of a pretrained network. The feature space of the network is required to be well-regularized using residual connections and spectral normalization. Feature densities serve as a proxy for the epistemic uncertainty of the model. To ensure a fair and meaningful comparison with DDU, we use the same pre-trained network as the base network for our repulsive ensemble head. We denote the unconstrained ensemble-head trained without repulsion loss as *MH-POVI* and the function space counterpart as *MH-f-POVI*.

**Uncertainty and evaluation metrics.** We quantify single network uncertainty using its *softmax entropy* $\mathbb{H}[p(\mathbf{y}|\mathbf{x}, \theta)]$, and *softmax density* (Liu et al., 2020b), i.e. the logsumexp of the logits. In ensemble models, we quantify aleatoric and epistemic uncertainty using the following decomposition

$$\underbrace{\mathbb{H}[\mathbb{E}_{p(\theta|\mathbf{X},\mathbf{Y})}[p(\mathbf{y}|\mathbf{x}, \theta)]]}_{\text{PREDICTIVE ENTROPY}} = \underbrace{\mathbb{E}_{p(\theta|\mathbf{X},\mathbf{Y})}[\mathbb{H}[p(\mathbf{y}|\mathbf{x}, \theta)]]}_{\text{ALEATORIC}} + \underbrace{\mathbb{I}[\mathbf{y}; \theta \mid \mathbf{x}, \mathbf{X}, \mathbf{Y}]}_{\text{EPISTEMIC}}$$

where the epistemic uncertainty is obtained by the mutual information between $\mathbf{y}$ and $\theta$ (Depeweg et al., 2018). The predictive entropy of the ensemble captures both aleatory and epistemic uncertainty. For estimating epistemic uncertainty using DDU, we compute the marginal likelihood of the GMM density $q(\mathbf{z}) = \sum_{\mathbf{y}} q(\mathbf{z}|\mathbf{y})q(\mathbf{y})$. This refers to as *GMM density* in the tables.

We evaluate the reliability of uncertainty predictions using two criteria: Calibration on in-distribution (ID) data and under-distribution shifts, and the ability to discriminate OOD data. For calibration, we use the expected calibration error (ECE) (Naeini et al., 2015), which measures the agreement between predicted uncertainties and the actual probabilities of accurate predictions. In addition, we report the area under the ROC curve (AUROC) for distinguishing correctly and incorrectly classified ID samples, referred to as InC and InW, respectively. For OOD detection, we evaluate the AUROC using the estimated epistemic uncertainty of the different methods.

## 5.1 UNCERTAINTY DECOMPOSITION

We test the ability of our model to discriminate between ambiguous ID and OOD data using the DirtyMNIST dataset (Mukhoti et al., 2023). This dataset contains artificially generated ambiguous inputs that belong to multiple classes. We use Fashion MNIST (Xiao et al., 2017) as OOD data. All methods are based on a Resnet-18 architecture with spectral normalization. For DDU and MH-f-POVI we use the feature space of the trained single model network. The MH-f-POVI ensemble head consists of 20 particles with two hidden layer of 20 neurons each and is trained for different sets of context points $\mathbf{x}_C$. Either we use a related dataset (KMNIST (Clanuwat et al., 2018)), random shuffling of image patches (PATCHES), or Gaussian noise (NOISE). We use a batch size of 64 for both training data and context points.

Fig. 3 shows the histograms of aleatoric versus epistemic uncertainty for MH-f-POVI, MH-POVI, and a deep ensemble. Without any constraints, MH-POVI (b) is unable to clearly discriminate between ambiguous and OOD data - both result in high aleatoric uncertainty. Including the function-space loss (see MH-f-POVI in (a)) increases the epistemic uncertainty for OOD data, while ambiguous data is reflected in the aleatoric uncertainty only. It illustrates that the multi-head architecture is sufficiently expressive, given that the optimization procedure is modified with the function space loss. Although not as strong, deep ensembles still reflect an increase in epistemic uncertainty for fashion MNIST compared to ambiguous MNIST.

Table 1 summarizes the ID performance and OOD detection gain we obtain by including the repulsive ensemble head for different choices of context points. Even without constraining the function

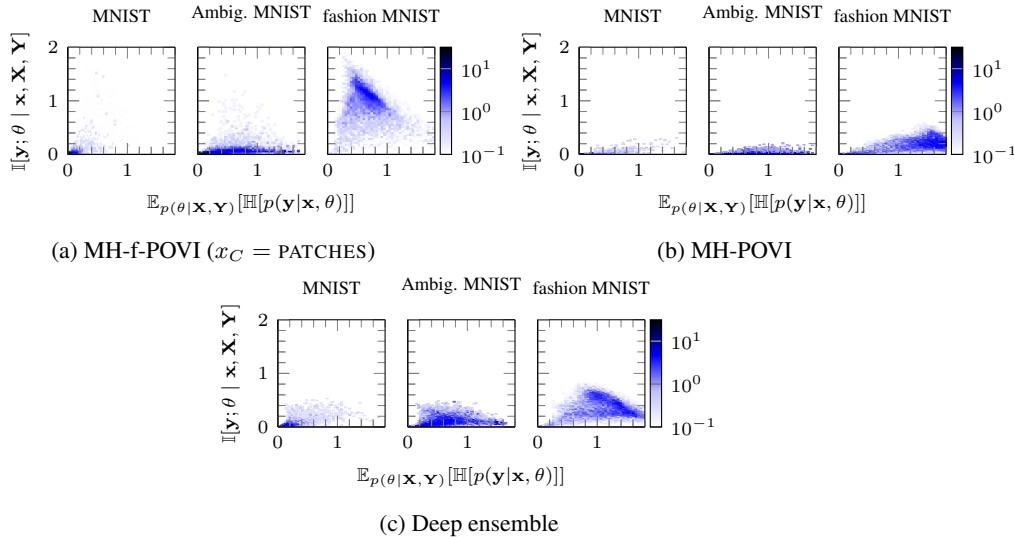

Figure 3: Histograms of aleatoric versus epistemic uncertainty on ID data (MNIST, Ambiguous MNIST) and OOD data (fashion-MNIST). We use an MH-f-POVI repulsion head on top of a pre-trained Resnet-18 with spectral normalization. We compare (a) MH-f-POVI with random patches as context points, (b) MH-POVI without a repulsion term, (c) and deep ensembles with 5 members.

space (MH-POVI), we were able to increase accuracy, improve calibration, and enhance OOD detection quality compared to the single network. Further improvement is achieved by incorporating the repulsion loss term (MH-f-POVI). By employing negative data augmentation on the training data through random slicing and shuffling of image patches (PATCHES), we could achieve competitive results to DDU and outperform deep ensembles. In terms of parameters, our MH-f-POVI ensemble head introduces the smallest increase with 2 % of additional parameters, while deep ensembles are 5 times the size of a single model.

In Appendix A.2 we follow the active learning experiment proposed in Mukhoti et al. (2023) where epistemic uncertainty estimates are used to iteratively select the most informative image samples from a pool of clean and ambiguous samples. We are able to achieve similar performance as DDU and deep ensembles, while being more parameter efficient.

Table 1: Comparison of uncertainty decomposition using different methods trained on DirtyMNIST. We evaluate ID accuracy (Acc.) and calibration (ECE) on MNIST. We report the AUROC for ID MNIST / ID Ambiguous MNIST and OOD fashion MNIST. Mean and standard deviation are computed over 20 runs. Best results are in bold, second best are underlined.

| | METHOD | ACC. (↑) | ECE (↓) | EPISTEMIC UNCERTAINTY | MNIST vs f-MNIST. AUROC (↑) | Ambig. vs f-MNIST AUROC (↑) | PARAM. (↓) |
|---|---|---|---|---|---|---|---|
| | Single model | 98.89% | 0.91% | Softmax Entropy Softmax Density | $98.42\%_{\pm 1.03}$ $98.75\%_{\pm 0.72}$ | $81.80\%_{\pm 7.01}$ $84.33\%_{\pm 5.55}$ | 100 % |
| | DDU | | | GMM Density | $99.61\%_{\pm 0.12}$ | $\mathbf{99.92\%_{\pm 0.03}}$ | 113.2 % |
| Dirty MNIST (ResNet-18) | MH-f-POVI (*ours*) ($\mathbf{x}_C$ = KMNIST) | 99.20% | 0.93% | Pred. Entropy Mutual Inf. | $\underline{99.76\%_{\pm 0.07}}$ $99.64\%_{\pm 0.10}$ | $97.84\%_{\pm 0.84}$ $\underline{99.52\%_{\pm 0.17}}$ | 102 % |
| | MH-f-POVI (*ours*) ($\mathbf{x}_C$ = PATCHES) | 99.26% | 0.88% | Pred. Entropy Mutual Inf. | $\mathbf{99.80\%_{\pm 0.06}}$ $99.68\%_{\pm 0.11}$ | $97.88\%_{\pm 0.82}$ $\underline{99.51\%_{\pm 0.20}}$ | |
| | MH-f-POVI (*ours*) ($\mathbf{x}_C$ = NOISE) | 99.19% | $\mathbf{0.60\%}$ | Pred. Entropy Mutual Inf. | $99.64\%_{\pm 0.14}$ $99.53\%_{\pm 0.17}$ | $94.13\%_{\pm 2.44}$ $98.41\%_{\pm 0.56}$ | |
| | MH-POVI (*ours*) | $\underline{99.32\%}$ | $\underline{0.63\%}$ | Pred. Entropy Mutual Inf. | $99.37\%_{\pm 0.50}$ $99.21\%_{\pm 0.36}$ | $90.53\%_{\pm 4.33}$ $96.49\%_{\pm 1.54}$ | |
| | 5-Ensemble | $\mathbf{99.37\%}$ | 1.65% | Pred. Entropy Mutual Inf. | $99.55\%_{\pm 0.16}$ $98.62\%_{\pm 0.33}$ | $92.13\%_{\pm 2.39}$ $92.02\%_{\pm 3.03}$ | 500 % |

Table 2: Comparison of various OOD detection methods for CIFAR10/CIFAR100. The parametric overhead of MH-f-POVI methods (*ours*) is minimal compared to alternatives while achieving competitive results on ID and OOD metrics. Mean and standard deviation are computed over 20 runs. Best results are in bold, second best are underlined.

| | Method | Acc. (↑) | ECE (↓) | Epistemic Uncertainty | InC vs SVHN AuROC (↑) | InC vs Tiny-Imagenet AuROC (↑) | Param. (↓) |
|---|---|---|---|---|---|---|---|
| **CIFAR10 (ResNet-18)** | Single model | 92.66% | 3.66% | Softmax Entropy
Softmax Density | $93.64\%_{\pm 2.32}$
$93.42\%_{\pm 2.84}$ | $89.25\%_{\pm 1.20}$
$88.18\%_{\pm 1.44}$ | 100% |
| | DDU | | | GMM Density | $90.95\%_{\pm 2.02}$ | $87.24\%_{\pm 0.71}$ | 113.2% |
| | MH-f-POVI (*ours*) ($x_C =$ CIFAR100) | 92.14% | 5.16% | Pred. Entr.
Mut. Inf. | $\underline{96.69\%}_{\pm 1.28}$
$95.99\%_{\pm 1.23}$ | $92.43\%_{\pm 0.30}$
$\underline{92.47\%}_{\pm 0.31}$ | 102% |
| | MH-f-POVI (*ours*) ($x_C =$ PATCHES) | 92.89% | $\underline{2.48}\%$ | Pred. Entr.
Mut. Inf. | $93.24\%_{\pm 2.58}$
$90.28\%_{\pm 2.91}$ | $91.26\%_{\pm 0.38}$
$90.52\%_{\pm 0.47}$ | |
| | MH-f-POVI (*ours*) ($x_C =$ NOISE) | $\underline{93.13}\%$ | 2.51% | Pred. Entr.
Mut. Inf. | $93.41\%_{\pm 3.08}$
$91.14\%_{\pm 2.67}$ | $90.13\%_{\pm 0.53}$
$89.75\%_{\pm 0.44}$ | |
| | MH-POVI (*ours*) | 93.07% | 3.26% | Pred. Entr.
Mut. Inf. | $94.58\%_{\pm 2.17}$
$92.88\%_{\pm 1.90}$ | $89.79\%_{\pm 0.67}$
$89.69\%_{\pm 0.61}$ | |
| | 5-Ensemble | **95.27%** | **1.99%** | Pred. Entr.
Mut. Inf. | $\mathbf{97.01\%}_{\pm 0.50}$
$94.52\%_{\pm 1.49}$ | $\mathbf{92.87\%}_{\pm 0.19}$
$91.03\%_{\pm 0.25}$ | 500% |
| **CIFAR100 (Wide-ResNet-28-10)** | Single model | $\underline{80.75}\%$ | 6.91% | Softmax Entropy
Softmax Density | $90.27\%_{\pm 1.32}$
$90.82\%_{\pm 1.41}$ | $88.81\%_{\pm 0.17}$
$87.91\%_{\pm 0.17}$ | 100% |
| | DDU | | | GMM Density | $77.51\%_{\pm 4.49}$ | $69.24\%_{\pm 2.24}$ | 157.4% |
| | MH-f-POVI (*ours*) ($x_C =$ CIFAR10) | 79.53% | 6.89% | Pred. Entr.
Mut. Inf. | $90.88\%_{\pm 1.41}$
$89.30\%_{\pm 1.46}$ | $\underline{89.19}\%_{\pm 0.21}$
$88.42\%_{\pm 0.26}$ | 100.9% |
| | MH-f-POVI (*ours*) ($x_C =$ PATCHES) | 79.82% | 5.71% | Pred. Entr.
Mut. Inf. | $\mathbf{92.28\%}_{\pm 1.46}$
$91.29\%_{\pm 1.31}$ | $88.56\%_{\pm 0.24}$
$88.51\%_{\pm 0.24}$ | |
| | MH-f-POVI (*ours*) ($x_C =$ NOISE) | 79.59% | 5.28% | Pred. Entr.
Mut. Inf. | $91.80\%_{\pm 1.49}$
$91.65\%_{\pm 1.25}$ | $88.75\%_{\pm 0.22}$
$88.21\%_{\pm 0.26}$ | |
| | MH-POVI (*ours*) | 79.66% | $\underline{4.75}\%$ | Pred. Entr.
Mut. Inf. | $91.52\%_{\pm 1.31}$
$91.30\%_{\pm 1.34}$ | $88.91\%_{\pm 0.24}$
$88.31\%_{\pm 0.20}$ | |
| | 5-Ensemble | **83.29%** | **1.76%** | Pred. Entr.
Mut. Inf. | $\underline{91.83}\%_{\pm 0.76}$
$88.82\%_{\pm 1.25}$ | $\mathbf{89.93\%}_{\pm 0.10}$
$86.97\%_{\pm 0.16}$ | 500% |

## 5.2 Semantic shift detection

We further test the OOD detection performance of our method on larger image classication tasks, summarized in Table 2. We train a Resnet-18 for CIFAR10 and a Wide-Resnet-28-10 for CIFAR100, both regularized using spectral normalization. We evaluate the AUROC between correctly classified ID data and incorrect ID data and OOD data combined, referred to as InC vs. OOD in Table 2. Specifically, we consider SVHN (as far OOD) and TinyImagenet (as near OOD). We argue that rejecting both unknown inputs and incorrect predictions is more relevant in real-world applications (Cen et al., 2023). In both tasks, our MH-f-POVI ensemble head consists of 20 particles with two hidden layers and 20 neurons per layer.

For CIFAR10, deep ensembles perform best in terms prediction accuracy and ECE calibration. However, MH-f-POVI with CIFAR100 as context points is able to match deep ensembles in detecting incorrect predictions and OOD data. Using shuffled image patches and noise as context points also leads to improvements over the single network and DDU in terms of accuracy and calibration. We repeated the same experiment for a Wide-Resnet-28-10 architecture, again observing improvements when including MH-f-POVI while introducing little parametric overhead compared to the other baselines (see Appendix A.3). For CIFAR100, the MH-f-POVI repulsion head introduces a slight decrease in accuracy. We argue that this occurs due to the training the repulsive head for a pre-trained network and a large number of classes, that leads to suboptimal solutions. Still, we achieve the best OOD detection for SVHN using MH-f-POVI with random patches as context points, even outperforming full deep ensembles. Due to the large number of classes, DDU requires significantly more parameters compared to our MH-f-POVI. While we obtained good results with DDU on the datasets with a smaller number of classes, we could not achieve a competive performance on CIFAR100. Again, the deep ensemble results in the best accuracy and calibration. However, we emphasize that

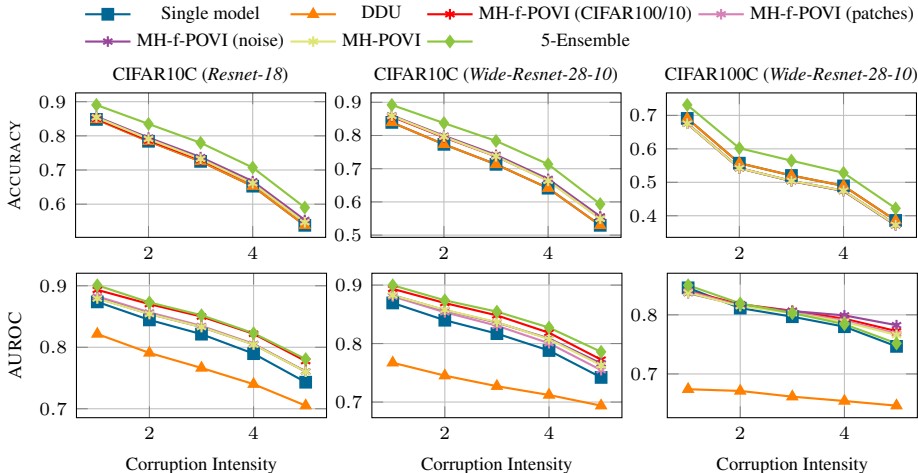

Figure 4: Accuracy and AUROC for single model, DDU, deep ensembles, and MH-f-POVI methods *(ours)* for different levels of corruption intensity, averaged over all corruption types. Deep ensembles achieve the highest accuracy *(top row)*. However, our method provides competitive uncertainty estimates for distinguishing between correct and incorrect predictions (AUROC, *bottom row*).

our aim is not to improve accuracy, but to introduce computationally cheap uncertainty estimates that are informative about incorrect predictions and OOD data.

## 5.3 COVARIATE SHIFT ROBUSTNESS

We analyze the behavior of our model when presented with corrupted data (CIFAR10C and CI-FAR100C (Hendrycks & Dietterich, 2019)) for two different network architectures, Resnet-18 and Wide-Resnet-28-10. Figure 4 shows the models accuracy and the AUROC between correct and incorrect predictions, averaged over all corruption types. We use softmax entropy for the single model, predictive entropy for deep ensembles and MH-f-POVI, and GMM density for DDU. Our MH-f-POVI method with context points from a related dataset performs competitively with deep ensembles for both network architectures and datasets, although the accuracy is not improved over the single model. Again, we emphasize that the goal of the MH-f-POVI is not necessarily to improve accuracy, but to provide calibrated uncertainty estimates at small additional computational cost. In contrast, the feature space density obtained by DDU is not a reliable indicator of the correctness of a prediction under covariate shift, resulting in the worst AUROC values, and deep ensembles come at substantial additional costs. This has been also observed in Postels et al. (2021), which show that uncertainty methods based on feature space density are consistently worse calibrated.

## 6 CONCLUSION

We have shown that particle optimization in function space is not limited to deep ensemble architectures. A significant number of parameters can be saved by exploring different network architectures to parameterize the function space. We proposed a hybrid approach using a multi-headed network. The shared base network acts as a feature extractor for the repulsive ensemble head. This offers a principled way to provide already trained networks with retrospective uncertainty estimates, and to incorporate prior functional knowledge into the training procedure. We empirically demonstrate that our method successfully disentangles aleatoric and epistemic uncertainty, detects out-of-distribution data, provides calibrated uncertainty estimates under distribution shifts, and performs well in active learning. At the same time, we significantly reduce the computational and storage requirements compared to deep ensembles.

For future work, we plan to further explore the trade-off between the multi-head and base network architecture and its impact on uncertainty estimates and prediction accuracy.

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

# A  APPENDIX

## A.1  TRAINING DETAILS

For particle-based inference in function space (Wang et al., 2019; D'Angelo & Fortuin, 2021), we relied on the implementation available at `https://github.com/ratschlab/repulsive_ensembles`, and for DDU (Mukhoti et al., 2023) at `https://github.com/omegafragger/DDU`. Table 3 summarizes relevant hyperparameters for training the base networks and the repulsive ensemble head.

Table 3: Implementation details and hyperparameter for the different experiments.

| TASK | ARCHITECTURE | HYPERPARAMETER | VALUE |
| --- | --- | --- | --- |
| IMAGE CLASSIFICATION BASE NETWORK | RESNET-18 WIDE-RESNET-28-10 | EPOCHS | 50 (DirtyMNIST) 300 (CIFAR10/100) |
| | | OPTIMIZER | SGD |
| | | LEARNING RATE | 0.1 0.01 (epoch 150) 0.001 (epoch 250) |
| | | MOMENTUM | 0.9 |
| | | SN COEFFICIENT | 3 |
| ACTIVE LEARNING BASE NETWORK | RESNET-18 | EPOCHS | 20 |
| | | OPTIMIZER | Adam |
| | | LEARNING RATE | 0.001 |
| MULTI-HEAD PARTICLES | FULLY CONNECTED | EPOCHS | 20 |
| | | # HIDDEN LAYER | 2 |
| | | # NEURONS PER LAYER | 20 |
| | | ACTIVATION | ReLU |
| | | LEARNING RATE | 0.01 |

## A.2  ACTIVE LEARNING

We evaluate the performance of our MH-f-POVI uncertainty estimates on an active learning task proposed in Mukhoti et al. (2023). Given a small number of initial training points and a large pool of unlabeled data, the aim is to select the most informative data points, which are subsequently used to retrain the network. We report the results using softmax entropy of a single Resnet-18, DDU density, mutual information (MI) of an ensemble of 3 Resnet-18s, and MH-f-POVI ($\mathbf{x}_C = \kappa$MNIST). We start with an initial training set of 20 samples and a pool of clean MNIST and ambiguous MNIST samples. The ratio of clean to ambiguous is 1:60. In each iteration, we include the 5 samples with the highest level of epistemic uncertainty. Fig. 5 shows that both DDU and MH-f-POVI are able to compete with deep ensembles. The results are averaged over 5 runs with different random seed.

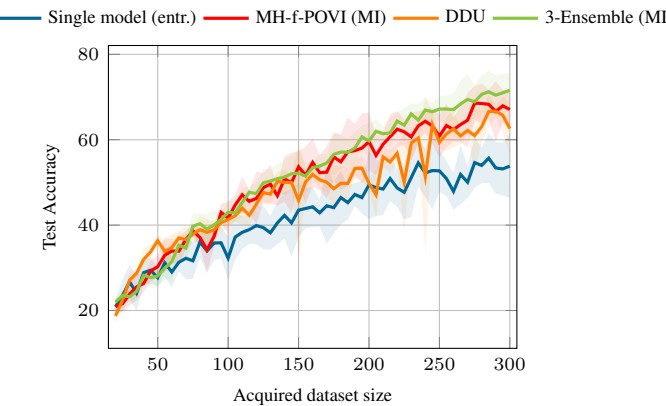

Figure 5: Test accuracy of the model as a function of the data samples that are acquired using the different uncertainty estimates. Using the mutual information (MI) of the MH-f-POVI prediction outperforms softmax entropy of the single network and performs on par with the other uncertainty baselines.

## A.3 INFLUENCE OF BASE NETWORK

Table 4: Influence of using a different base network architecture on CIFAR10. In both cases, our MH-f-POVI multi-head architecture consistently improves the single-network baseline and can compete with DDU and deep ensembles, with minimal parameter overhead compared to the alternatives. Best results are in bold, second best are underlined.

| | METHOD | ACC. (↑) | ECE (↓) | EPISTEMIC UNCERTAINTY | InC vs SVHN AuROC (↑) | InC vs Tiny-Imagenet AuROC (↑) | PARAM. |
|---|---|---|---|---|---|---|---|
| **ResNet-18** | Single model | 92.66% | 3.66% | Softmax Entropy | $93.64\%_{\pm 2.32}$ | $89.25\%_{\pm 1.20}$ | 100 % |
| | | | | Softmax Density | $93.42\%_{\pm 2.84}$ | $88.18\%_{\pm 1.44}$ | |
| | DDU | | | GMM Density | $90.95\%_{\pm 2.02}$ | $87.24\%_{\pm 0.71}$ | 113.2 % |
| | MH-f-POVI (*ours*) ($x_C$ = CIFAR100) | 92.14% | 5.16% | Pred. Entr. | $\underline{96.69}\%_{\pm 1.28}$ | $92.43\%_{\pm 0.30}$ | |
| | | | | Mut. Inf. | $95.99\%_{\pm 1.23}$ | $\underline{92.47}\%_{\pm 0.31}$ | |
| | MH-f-POVI (*ours*) ($x_C$ = PATCHES) | 92.89% | $\underline{2.48}$% | Pred. Entr. | $93.24\%_{\pm 2.58}$ | $91.26\%_{\pm 0.38}$ | 102 % |
| | | | | Mut. Inf. | $90.28\%_{\pm 2.91}$ | $90.52\%_{\pm 0.47}$ | |
| | MH-f-POVI (*ours*) ($x_C$ = NOISE) | $\underline{93.13}$% | 2.51% | Pred. Entr. | $93.41\%_{\pm 3.08}$ | $90.13\%_{\pm 0.53}$ | |
| | | | | Mut. Inf. | $91.14\%_{\pm 2.67}$ | $89.75\%_{\pm 0.44}$ | |
| | MH-POVI (*ours*) | 93.07% | 3.26% | Pred. Entr. | $94.58\%_{\pm 2.17}$ | $89.79\%_{\pm 0.67}$ | |
| | | | | Mut. Inf. | $92.88\%_{\pm 1.90}$ | $89.69\%_{\pm 0.61}$ | |
| | 5-Ensemble | **95.27**% | **1.99**% | Pred. Entr. | $\mathbf{97.01}\%_{\pm 0.50}$ | $\mathbf{92.87}\%_{\pm 0.19}$ | 500 % |
| | | | | Mut. Inf. | $94.52\%_{\pm 1.49}$ | $91.03\%_{\pm 0.25}$ | |
| **Wide-Resnet-28-10** | Single model | 92.20% | 4.04% | Softmax Entropy | $92.07\%_{\pm 4.52}$ | $88.34\%_{\pm 1.50}$ | 100 % |
| | | | | Softmax Density | $92.96\%_{\pm 4.67}$ | $87.44\%_{\pm 2.02}$ | |
| | DDU | | | GMM Density | $\underline{96.32}\%_{\pm 0.91}$ | $82.69\%_{\pm 1.34}$ | 106.25 % |
| | MH-f-POVI (*ours*) ($x_C$ = CIFAR100) | 93.31% | $\underline{2.12}$% | Pred. Entr. | $95.32\%_{\pm 3.24}$ | $91.68\%_{\pm 0.53}$ | |
| | | | | Mut. Inf. | $94.90\%_{\pm 3.03}$ | $\underline{91.73}\%_{\pm 0.49}$ | |
| | MH-f-POVI (*ours*) ($x_C$ = PATCHES) | 93.49% | **1.95**% | Pred. Entr. | $92.90\%_{\pm 3.65}$ | $91.09\%_{\pm 0.71}$ | 100.74 % |
| | | | | Mut. Inf. | $90.39\%_{\pm 3.73}$ | $90.53\%_{\pm 0.73}$ | |
| | MH-f-POVI (*ours*) ($x_C$ = NOISE) | $\underline{93.51}$% | 2.74% | Pred. Entr. | $94.30\%_{\pm 3.65}$ | $89.89\%_{\pm 1.12}$ | |
| | | | | Mut. Inf. | $92.39\%_{\pm 3.55}$ | $89.75\%_{\pm 0.44}$ | |
| | MH-POVI (*ours*) | 93.48% | 3.11% | Pred. Entr. | $97.76\%_{\pm 0.63}$ | $89.79\%_{\pm 0.67}$ | |
| | | | | Mut. Inf. | $95.47\%_{\pm 1.18}$ | $89.79\%_{\pm 0.97}$ | |
| | 5-Ensemble | **95.47**% | 2.76% | Pred. Entr. | $\mathbf{97.76}\%_{\pm 0.63}$ | $\mathbf{92.86}\%_{\pm 0.23}$ | 500 % |
| | | | | Mut. Inf. | $95.47\%_{\pm 1.18}$ | $91.15\%_{\pm 0.35}$ | |

### A.4 Base network architecture

We apply the repulsive ensemble head in a post-hoc fashion on top of a trained deep feature extractor. If the underlying network is unable to extract features that are useful for distinguishing OOD data from training data, enforcing diversity on such context points will result in reduced ID performance. Various related works use spectral normalization to regularize the feature space of the base network (Mukhoti et al., 2023; Liu et al., 2020a). To preserve distance awareness within a network with residual connections, one method is to constrain the spectral norm of the network weights. This constraint serves to regulate the bi-Lipschitz constant of the neural network.

First, we examine the impact of the choice of underlying feature extractor by comparing different base network architectures, namely LeNet-5, Resnet-18, and VGG-16 (in a first step without any feature space regularization). We evaluate the performance of uncertainty decomposition for the Dirty MNIST task as detailed in Section 5.1. The results for ID accuracy and OOD detection are averaged over 10 seeds and shown in Figure 6. We report the ID accuracy and OOD detection performance for clean MNIST and ambiguous MNIST data points. Both single model and deep ensemble are suitable to distinguish between clean training data and OOD samples for all network architectures. Considering ambiguous data, however, OOD detection deteriorates due to insufficient uncertainty decomposition between aleatoric (ambiguous data) and epistemic (OOD data) uncertainty. As reported in the original paper (Mukhoti et al., 2023), DDU relies on a well regularized feature extractor and significantly underperforms for the LeNet-5 architecture. Given a Resnet-18 architecture, on the other hand, DDU can reliably improve the distinction between ambiguous input samples and OOD samples. We are examining our repulsive ensemble head, MH-f-POVI, which consists of 20 ensemble members (particles), each with a linear layer. We evaluate its behavior for different choices of context points (i.e., kMNIST, patches, noise). The ensemble head is shown to improve the distinction between ambiguous and OOD data for all network architectures compared to the single-network baseline and the full deep ensemble. The largest improvement is observed for the use of informative context points (kMNIST), followed by the permutation of random image patches. Training the ensemble head without added constraints does not yield notable gains.

Next, we examine the sensitivity in terms of the spectral normalization strength of a Resnet-18, shown in Figure 7 for image classification and OOD detection using CIFAR10. We do not observe a strong correlation between the spectral normalization coefficient and the OOD detection performance for any method. Similar results have been reported in Postels et al. (2021).

### A.5 Ensemble head architecture

In the following, we test the sensitivity of the network size for the repulsive ensemble heads.

We begin with the two moons toy example, as illustrated in Figure 8, and demonstrate that non-linear decision boundaries are necessary for the ensemble head to achieve diverse predictions far from the training data. The base network consists of three fully connected hidden layers with 128 neurons. There is no feature collapse in this scenario – the input data is merely transformed to be linearly separable. In this case, nonlinear decision boundaries allow for uncertainty estimates that reflect the underlying data distribution.

Contrary to intuition, a repulsive ensemble consisting of linear layers performs well in the more complex task of OOD detection in image classification, without the need for additional nonlinearities. For the image classification task, we use a Resnet-18 as the underlying feature extractor. The use of informative context points enables repulsive linear layers to significantly improve OOD detection. Enlarging the network size of the ensemble heads does not necessarily result in further improvements. The base network is able to extract useful features for CIFAR10, but we still require informative context points to properly construct the linear classification layers, as shown in Figure 9, where the use of CIFAR100 as context points performs comparably to full deep ensembles. Alternative context point choices do not noticeably improve the base network.

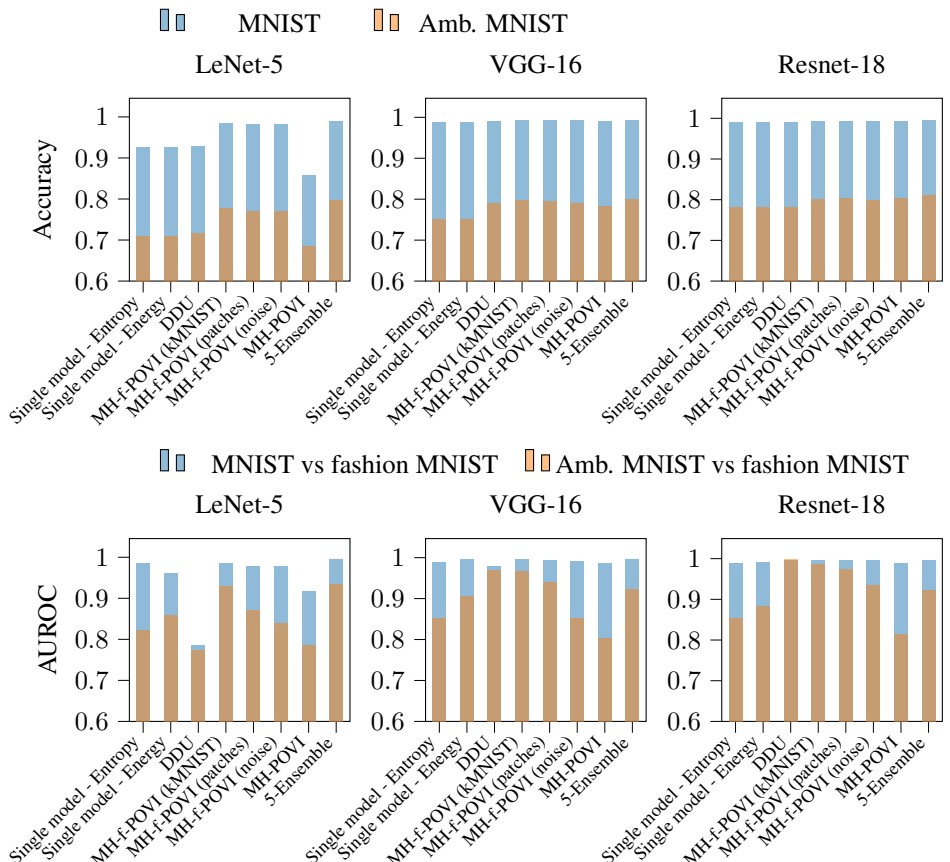

Figure 6: ID accuracy and OOD detection on Dirty MNIST (MNIST + ambiguous MNIST) using different uncertainty methods for varying base network architectures.

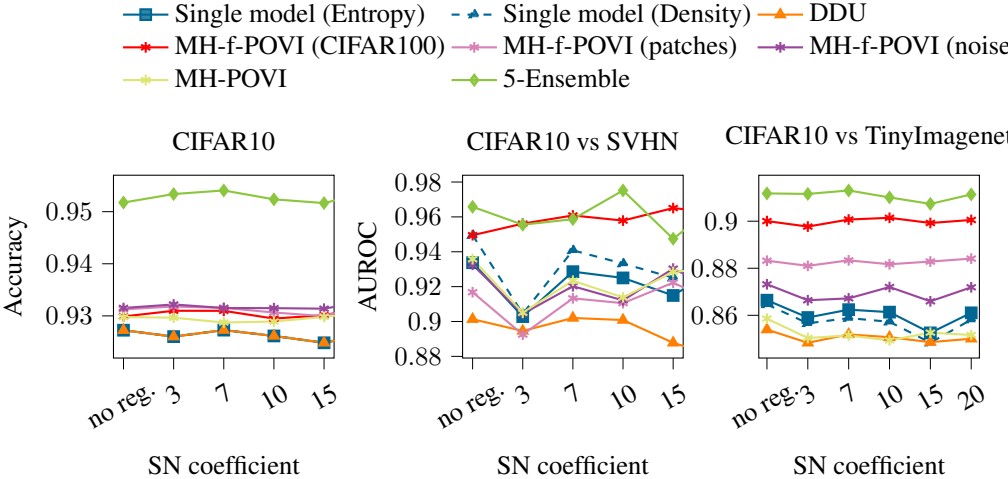

Figure 7: Influence of the spectral normalization coefficient on the ID and OOD performance of different uncertainty methods.

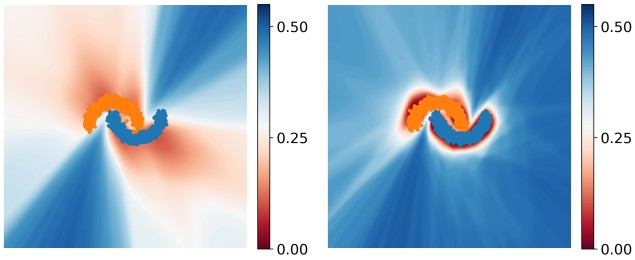

Figure 8: Uncertainty estimated using MH-f-POVI with an ensemble head consisting of 30 linear classifiers *(left)*, and an ensemble head with 2x20 network size *(right)*. Nonlinear decision boundaries are necessary for the uncertainty to represent the data density.

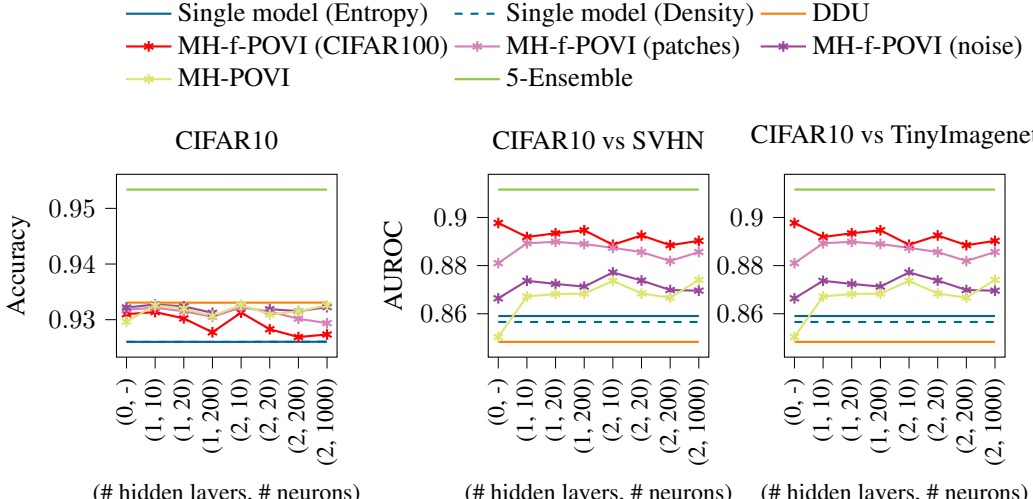

Figure 9: Influence of the network size of the MH-f-POVI and MH-POVI methods. Contrary to the 2-dimensional toy example, an ensemble of linear classifiers performs well, given appropriate context points for the function-space methods.

## A.6 NUMBER OF PARTICLES

Lastly, we examine the number of particles, representing the number of ensemble head members (see Figure 10). We use a Resnet-18 as a base model and a repulsive ensemble head consisting of linear layers. Again, the best performance of MH-f-POVI is obtained by using unlabeled CIFAR100 samples as context points. We notice an initial rise in OOD detection as more particles are introduced, followed by a drop in ID accuracy and OOD detection. As the number of particles notably increases, the repulsion term prevails over the likelihood term in the loss function. Enforcing excessive diversity on context points that capture training data features results in decreased ID accuracy (see Figure 10 for context points CIFAR100, and patches). Imposing diversity on random noise, on the other hand, has no effect on test accuracy. By reducing the strength of the repulsion term in the loss function through a scaling parameter (Figure 11), we avoid the loss of ID accuracy. We show that for 5, 20, and 100 particles a decreasing weight of the repulsion term compared to the likelihood term mitigates ID performance degradation and improves OOD detection.

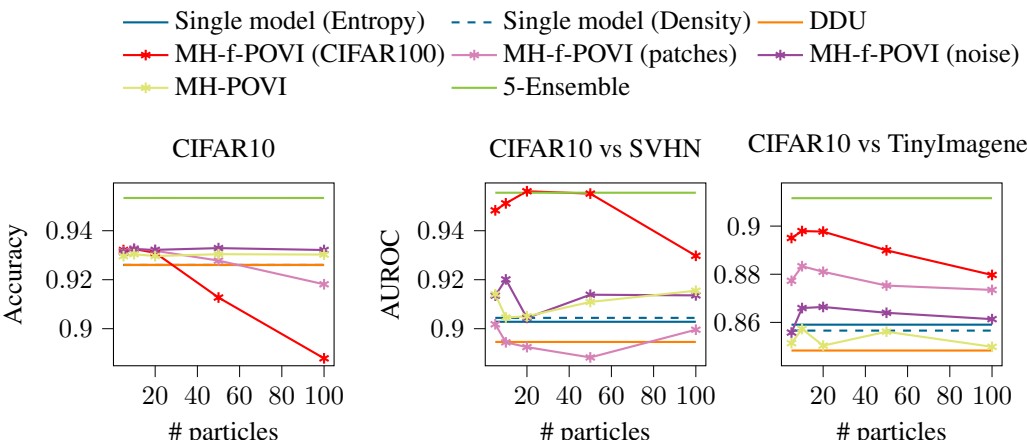

Figure 10: Influence of the number of particles, while keeping the scaling of the repulsion loss constant. A larger number of particles leads to a decrease in ID accuracy and OOD detection.

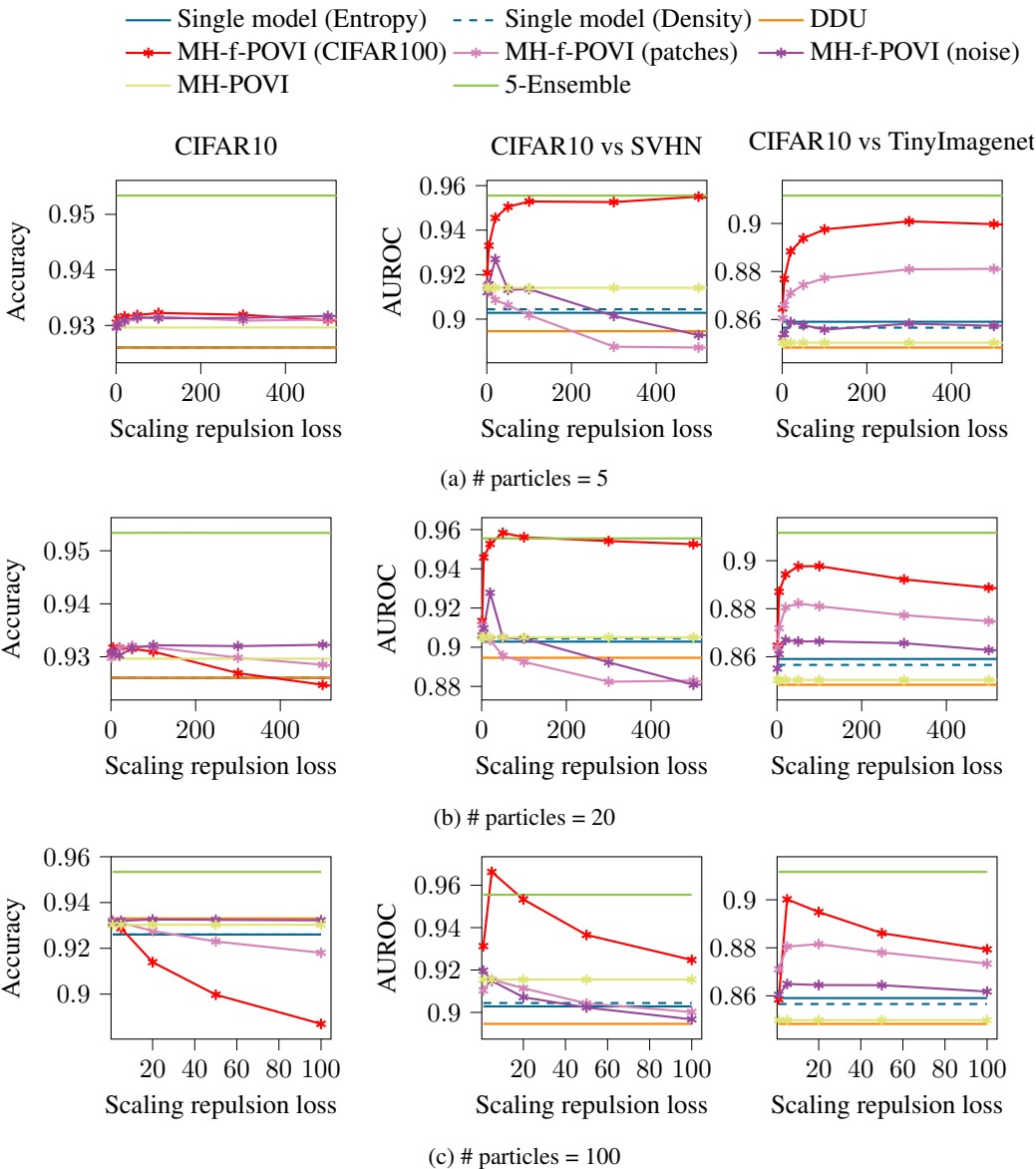

Figure 11: Influence of the scaling of the repulsion loss term for an ensemble head with (a) 5, (b) 20, and (c) 100 particles for ID accuracy and OOD detection.

## A.7 CHOICE OF CONTEXT POINTS

The main take-away from our ablation experiments is the sensitivity of the choice of context points. This influences results considerably more than the network size of the ensemble head or number of particles. The best results are obtained by leveraging unlabeled data points to enforce diverse predictions by the ensemble heads. Similarly, Kristiadi et al. (2022); Rudner et al. (2023) include OOD data points to Bayesian methods to improve OOD detection. Hendrycks et al. (2019) introduce an additional loss function to enforce the softmax scores of a single network to be uniformly distributed on a given OOD dataset. In our work, we demonstrate an additional benefit of enforcing diversity between the individual ensemble heads for uncertainty decomposition: Ambiguous ID data points lead to ensemble heads that *agree to disagree*, i.e. they all produce a similar distribution of the softmax score that reflects the ambiguity of the input data. Whereas on unseen OOD data, we enforce the individual models to return diverse predictions that reflect the missing knowledge. We demonstrated that in most cases the base network trained solely on ID data is capable of extracting sufficiently informative features for detecting OOD data – however, it is necessary to utilize informative context points.

