# OpenReview forum: "Lightweight uncertainty modelling using function space particle optimization"
_ICLR.cc/2024/Conference — Submitted to ICLR 2024_

### Official Review · Reviewer_f6Ta · 2023-10-23

**Soundness:** 2 fair
**Presentation:** 3 good
**Contribution:** 1 poor
**Rating:** 3
**Confidence:** 4

**Summary:**

This paper proposes a simple and efficient method for uncertainty estimation, namely to use a single deterministic network backbone with a small multi-head classifcation/regression head that is trained via a particle optimization objective that ensures diversity. This allows for a parametrically and computationally efficient alternative to ensembles. The experiments report improved performance over single deterministic networks and DDU on a range of in- and out-of-distribution uncertainty estimation tasks on various vision classification datasets.

Overall, while the work achieves its aim of introducing a computationally efficient method, the paper does not compare to any competitive baselines despite a range of comparable methods having been introduced in recent years. In conjunction with no new technical material being introduced, I think the paper should be rejected.

**Strengths:**

- the approach is described and motivated clearly
- the method is efficient and pragmatic
- paper is well-referenced, prior work that the paper builds upon is credit and the paper is properly contextualized in the literature

**Weaknesses:**

- most importantly, the paper sorely lacks a competitive baseline for efficient uncertainty estimation. DDU underperforms even the single net baseline on all non-MNIST benchmarks, on some of them by a large margin (CIFAR100 in Table 2, bottom panel of Fig 4). I don't know if DDU hasn't been tuned properly or simply isn't suited to tasks with a large number of classes, but in either case there are plenty of alternatives available in the literature that have reported good results in this setting, e.g. (Liu et al., Simple and Principled Uncertainty Estimation with Deterministic Deep Learning via Distance Awareness, Neurips 2020) or (Kristiadi et al., Posterior Refinement Improves Sample Efficiency in Bayesian Neural Networks, 2022).
- the paper is a bit light on ablations. In particular the use of the spectral normalization is not justified empirically. From a theoretical standpoint, as far as I am aware, prior work has justified its need based on the network being distance preserving being necessary for the use of a distance aware output head, however I do not see the proposed method relying on this property, so I would like to see a baseline without spectral normalization. Similarly, the structure of the ensemble heads appears as a magic hyperparameter without justification. I realize that a linear head without hidden layers will probably underperform due to the categorical log likelihood being a convex/unimodal objective w.r.t. the weights of the linear output layer, but this should be justified explicitly and sensitivity of the method w.r.t. this hyperparameter evaluated.
- the paper does not introduce any new technical material. This is in itself of course not grounds for rejection, combining prior work can be valuable, however I don't see any technical or theoretical contributions that would make up for the lacking empirical evaluation.

**Questions:**

To summarize the weaknesses, I would suggest to:
- add a couple of competitive(!) baselines from the literature on efficient uncertainty estimation and evidential deep learning
- justify the use of spectral normalization via an ablation study
- evaluate the sensitivity of the method w.r.t. depth/width of the ensemble head and discuss the trade-off vs. computational efficiency

---

> ### Author Response · Authors · 2023-11-22
>
> Dear Reviewer,
>
> We appreciate your evaluation and thank you for your feedback.
>
>
> We acknowledge the importance of competitive baselines and recognize that the absence of alternatives in our current version is a limitation. Due to time limitations we could not include them in this submission. However, we added ablation studies with experimental results for additional base network architectures and varying spectral normalization coefficients. We show that our method improves OOD detection performance for all analyzed base network architectures. We observe the best performance for the Resnet-18 architecture, although no strong correlation between spectral normalization strength and OOD detection performance. The appropriate choice of context points is still the most important factor in all experiments. The key point of this this work is that the use of the function space variational inference method with an appropriate choice of context points can significantly improve the performance of a single network.
>
> The additional ablation studies in the appendix are structured as follows:
>
> A4) Base network architecture
>
> A5) Ensemble head architecture
>
> A6) Number of particles
>
> A7) Choice of context points
>
>
> We genuinely appreciate your feedback and the opportunity to improve our paper. Thank you for your time and consideration.

---

### Official Review · Reviewer_bAmy · 2023-10-31

**Soundness:** 3 good
**Presentation:** 3 good
**Contribution:** 3 good
**Rating:** 6
**Confidence:** 5

**Summary:**

This paper is about uncertainty estimation with function space particle optimization. Its basically an ensemble, where each ensemble member represents one particle, and a special loss function repels particles away from each other, to ensure diversity in the ensemble. The authors propose to use a lightweight network architecture that uses a shared representation network and multiple prediction heads that each represent a particle.

Contributions are
- A method for uncertainty estimation using a single network architecture with multiple heads and function-space particle optimization, which is more computation and parameter efficient than a full ensemble.
- Results that show that the particle optimization in ensemble heads provides high quality uncertainty estimation in various settings including out of distribution detection and calibration, assuming the right regularization.
- Results that show successful disentangling of aleatoric and epistemic uncertainty on MNIST for active learning, and near and far out of distribution detection on CIFAR10, and provide good uncertainties under distribution shift (corruptions).

**Strengths:**

- The paper is mostly well written and kind of easy to understand. I have reservations below about some details of the model.
- The idea makes sense, to increase diversity of ensemble heads by treating the like particles that repel to each other and explore different parts of the parameter/function space, this seems to be applied to ensembles and now the authors propose to use a ensemble with a shared representation network, which lowers the computation costs for inference time. This is important due to researchers and practitioners not using uncertainty estimation methods because of increased computational costs.
- The selection of baselines seems to be appropriate, even as I make suggestions for better baselines in the minor comments. The paper compares against DDU and ensembles, DDU is a good and recent baseline for lightweight uncertainty estimation (using a single model), while ensembles is the state of the art for simple ways to obtain high quality uncertainty estimation.
- The evaluation seems to be correct, the paper uses Dirty MNIST and CIFAR10/100 vs several out of distribution datasets, including out of distribution classification of incorrect predictions, and evaluates appropriate metrics: accuracy, expected calibration error, and AUROC for OOD detection.
- Results indicate that the propsed method MH-POVI and MH-f-POVI performs closely to a ensemble and sometimes it outperforms DDU (also a single model) in terms of accuracy, calibration error, and out of distribution performance, often coming in 2nd place behind ensembles. I believe these results show that particle optimization for shared network ensembling is a viable strategy.
- There is a good set of ablation results, varying the selection of context points (three) for the functional variant of MH-POVI, showing the effect of this parameter on performance, and also OOD detection on corrupted versions of CIFAR10 and 100, showing that MH-POVI and functional variant are closer to ensembles in performance loss than it is to DDU (which performs worse).

**Weaknesses:**

- I have doubts about the fairness of the comparisons, as 20-30 particles are used for MH-POVI and variations, but only an ensemble of five networks. I believe a fairer comparison is to equalize the number of ensemble networks and particles, or to evaluate both methods with a variable number of particles and ensemble members, which also would provide information about how both methods scale with the number of particles.

~~- To me it is not clear what is the form of the loss function that is used to train the model. Equation 2 provides the update rule in the function space, and the paper argues that function particles are represented as neural networks, but then Equation 2 seems to use Equation 1 (in particular the v(f^i) ), but Equation 1 is for parameter-space, and to me it is not clear how to generalize the update rule to function space.~~

Minor Comments
- About multi-head architectures for uncertainty, there is also the concept of a sub-ensemble, which is very similar to the proposed method in this paper minus the particle optimization, it is meant to be an approximation to a deep ensemble, here I do not suggest a comparison but just integration into the literature for multi-head network architectures.

Valdenegro-Toro, M., 2023. Sub-Ensembles for Fast Uncertainty Estimation in Neural Networks. In Proceedings of the IEEE/CVF International Conference on Computer Vision (pp. 4119-4127).

~~- In disentangled classification uncertainty, the authors use a decomposition of predictive entropy using the softmax entropy (aleatoric) and the mutual information (epistemic), could you include information or clarify, how the mutual information is computed for the ensemble heads?~~

~~- Another suggestion for baselines, is to compare against more lightweight uncertainty quantification methods (currently only DDU), for example DUQ, Evidential Deep Learning, etc.~~

- In Figure 3, I believe the paper has to say how to interpret these plots, that in the OOD setting ideally aleatoric uncertainty should be low and epistemic uncertainty should be high, while the in-distribution setting it is the opposite (epistemic low and aleatoric variable depending on samples).
- In Section 3, please explain what is negative data augmentation, this term is not clear from the context of the paper and no citation for further information is provided.

**Questions:**

- Could you clarify how the network heads are trained using function space particle optimization? In particular how Eq 2 is evaluated.
- Is the model trained end-to-end? Please clarify the training process.
- Can you motivate the selection of context points? For example using Gaussian noise as a context point, how do you know that these points are supported by the data distribution?

---

> ### Author Response · Authors · 2023-11-22
>
> Dear Reviewer,
>
> We greatly appreciate your careful review of our paper.
>
>
> Thank you for bringing the work of Valdenegro-Toro (2023) to our attention. In the revised manuscript, we will incorporate a reference to this work in the related works section.
>
> Regarding your doubts towards the fairness of the comparison: We compared with a deep ensemble consisting of 5 ensemble members in order to keep the number of additional parameters comparable. We added additional ablation studies (see appendix A5, A6), where we evaluate the influence of the ensemble head network architecture and the number of particles. There, we show that for the image classification OOD detection task an ensemble of linear layers can already achieve competitive results. However, this is only achieved for an appropriate selection of context points. By using linear layers, we only add minimal computational and memory costs for the forward pass. Thus, we argue that the comparison to the 5-member deep ensemble is fair, given the significant reduction in parameter cost. However, in our additional experiment we show the results of our ensemble head consisting of 5 particles (see Figure 11) to address your concern.
>
> The network heads with the function space particle optimization are trained as follows:
> In each training iteration we load a batch of training data and context points. The context points are used to approximate the particles in function space. For these context points, the repulsion kernel (in our case a Gaussian kernel) enforces diverse predictions of each ensemble member. The other objective, the posterior term, enforces that all particles correctly predict the training data.
>
> We train our model post-hoc without changing the base network in order to maintain a fair comparison to DDU and the base network scores. Our key observation is the importance of context points for the uncertainty estimation performance (see appendix A5-A7). In all added ablation studies, we test our method for the same selection of context points as in the main paper. In each experiment we make the same observation: The largest improvements are obtained by using unlabeled data points that are related to the training data. As you correctly mentioned, using Gaussian noise does not improve the uncertainty estimates of the function space method compared to the unconstrained ensemble head. This highlights the need and usefulness of context points that carry additional information - in this case, the post-hoc extension to the single model can be significantly improved.
>
>
> Thank you once again for your thorough review and valuable feedback. We appreciate your time and effort.

---

### Official Review · Reviewer_y234 · 2023-11-01

**Soundness:** 2 fair
**Presentation:** 2 fair
**Contribution:** 2 fair
**Rating:** 3
**Confidence:** 5

**Summary:**

Summary:
This article presents a lightweight approach to uncertainty modeling that provides calibrated
uncertainty estimates by utilizing particle-based variational inference in function spaces. Unlike
methods that use deep ensemble representations of particles, this approach presents a multi-headed
neural network that achieves a reduction in computational requirements. By sharing a joint latent
representation, the computational requirements are reduced, while the multi-head network maintains
the diversity of predictions.

**Strengths:**

Strengths:
1) This article presents a method for uncertainty estimation based on particle inference in function
space, along with a hybrid method using multi-head networks.
2) Using multi-headed networks instead of full deep integration methods for lightweight purposes.
3) The article examines out-of-distribution data, provides uncertainty estimates for calibration
under distributional shifts, and obtains certain experimental results.

**Weaknesses:**

1)Given the emphasis on 'LIGHTWEIGHT' in the article's title, I expect to see more in-depth
analysis and explanation of this theme in the main text to help readers better understand its
role in the research.
2)The authors evaluated the proposed multi-headed architecture on several benchmark tasks.
However, the reasons for the selection of these benchmark tasks were not discussed.
3) The topic of the article is uncertainty estimation under lightweight, the author's description of
lightweight is not sufficient, and the analysis of lightweight is lacking in the experimental part.
 4) The author's contribution is not enough, the multi-head network and particle variational
optimization methods used are directly from other places, and the article lacks a certain degree
of innovation.

**Questions:**

N/A

---

> ### Author Response · Authors · 2023-11-22
>
> Dear Reviewer,
>
> we appreciate your thoughtful review on our paper. We have carefully considered your review, and we would like to share the revisions and clarifications we have made in response to your comments.
>
> 1) We provide additional ablation studies in the appendix of the revised version. The additional ablation studies in the appendix are structured as follows:
>
> A4) Base network architecture
>
> A5) Ensemble head architecture
>
> A6) Number of particles
>
> A7) Choice of context points
>
> We show that for the image classification task, a repulsive ensemble of linear layers is sufficient to improve OOD detection given that the context points used for the repulsion term are appropriate and informative. By using unlabeled and related image data (kMNIST for DirtyMNIST, CIFAR100 for CIFAR10), we can improve the single model by a large margin with minimal computational overhead for the forward pass.
>
> 2) The experimental setting has been selected to be similar to cited related works to simplify comparison.
>
> 3) We show that we are able to improve the uncertainty estimates of a single network by adding only a minimum number of additional parameters. During the training of the ensemble head, we use the same batch size for training data and context points. By relying on the features extracted from the pretrained base network, we show in our additional experiments that the size of the ensemble head can be minimal and still provide informative uncertainty estimates.
>
> We hope that the additional experiments and discussions contribute to a better understanding of our proposed model.

---

### Official Review · Reviewer_Runa · 2023-11-02

**Soundness:** 2 fair
**Presentation:** 3 good
**Contribution:** 2 fair
**Rating:** 5
**Confidence:** 4

**Summary:**

In this work, the authors proposed a hybrid approach using a multi-head network to achieve lightweight uncertainty modeling. The method provides calibrated uncertainty and also preserves bi-Lipschitz conditions by leveraging particle-based variational inference in function space. They also claim that the method achieves competitive results in disentangling aleatoric and epistemic uncertainty for multiple UQ tasks, including active learning, OOD detection, and distribution shifts with reduced compute and memory requirements.

**Strengths:**

1. The paper is well-written and easy to follow with strong related work and background.
2. The topic is very interesting and aims to address the current bottleneck of deep ensemble and DUQ studies.
3. The experiments demonstrate the effectiveness of the proposed method clearly and provide a comprehensive assessment of multiple tasks in the uncertainty estimation area.

**Weaknesses:**

1. The novelty contribution seems not very strong. Most of the core components in this framework all exist while the development of the framework is a nontrivial task.
2. Beyond the multi-head architectures, we did not directly capture the core contribution of the new method compared with the existing deep ensembles or DUQ methods.
3. The method section lacks enough details and discussions such that it might not be easier to reproduce the method.
4. Some key claims, like disentangling uncertainties and preserving bi-Lipschitz conditions are not well supported from the theoretical perspective.  We only observed improved empirical results but may not fully understand why the proposed methods bring such advantages

**Questions:**

1. How to preserve the bi-Lipschitz conditions and how to avoid feature collapse?  What's the key component to deal with these challenges? Can you show some ablation studies to verify the effectiveness?

2. How to disentangle aleatoric and epistemic uncertainty?  Which features mainly handle these capabilities？ What's the theoretical foundation of the proposed method or by leveraging existing works?

3. How about the computational cost and memory cost comparison with the baseline methods on multiple tasks?  I think it is important to show these since the paper mainly claims the high efficiency and low memory requirements.

---

> ### Author Response · Authors · 2023-11-22
>
> Dear Reviewer,
>
> Thank you for your detailed review and we appreciate the opportunity to address these aspects of our work. Below are our responses to each of your questions:
>
>
> We used the uncertainty decomposition proposed in Depeweg, Stefan, et al. "Decomposition of uncertainty in Bayesian deep learning for efficient and risk-sensitive learning." to distinguish between aleatoric and epistemic uncertainty. The equation is provided on page 6.
>
> Additionally, we added ablation studies regarding the influence of the base network and the use of spectral normalization for networks with residual connections (see appendix A4 of the revised version). We show that our method is not as sensitive to the choice of architecture and spectral normalization strength, *given* an appropriate choice of context points. We also show that in the image classification task, an ensemble of repulsive linear layers can already achieve competitive results to deep ensembles (see appendix A5). By using informative context points, e.g. unlabeled and related image data (kMNIST for DirtyMNIST, CIFAR100 for CIFAR10), we can improve the single model by a large margin with minimal computational overhead for the forward pass.
> Regarding the choice of number of particles we added an ablation study (appendix A6). E.g, for CIFAR10 we can achieve competitive results when using 20 linear layers on top of the pre-trained feature space, which is a small overhead in terms of computational and memory costs compared to the Resnet-18 base network.
>
>
> We highly appreciate your time and consideration in reviewing this paper and hope that our comments and ablation studies could provide some clarification.

---

> > ### Comment · Reviewer_Runa · 2023-11-23
> > **Thanks for your response**
> >
> > Dear author team,
> >
> > Thanks for your response, which is helpful to better understand the method and experiments.

---

### Official Review · Reviewer_KBVV · 2023-11-03

**Soundness:** 3 good
**Presentation:** 3 good
**Contribution:** 2 fair
**Rating:** 3
**Confidence:** 4

**Summary:**

The authors propose a functional uncertainty quantification technique for neural networks based on particle representations which has the following core innovation: it attempts to share a backbone architecture and replace heads per particle.

Taking things from the top, the authors embrace the particle-based VI literature in a functional view and combine it with spectral normalization to obtain smooth "head" representations and capture densities better.

They largely follow the path forged by D'Angelo and Fortuin, with minor modifications int he shared backbone and addition of spectral normalization which was explored heavily in DDU and other works that they attribute correctly.

In practice, the functional view trades off parameters for memory and data-constraints and becomes quasi-semi-parametric, so the authors also consider ways to generate useful "context points" to evaluate their functions so that the function-space kernel works correctly.

The performance of the technique seems fine evaluated over a variety of tasks where authors capture accuracy, in-distribution, and out-of-distribution metrics combined with parameter count.

**Strengths:**

The paper proposes a combination of known techniques and architecture changes that ultimately works.

The idea of sharing backbones is sound and has been applied to ensembles before, and combining their heads with spectral normalization to create better density models is also a reasonable step aligned with many successful papers.

Finally, the experiments indicate that their setup works well quantitatively with less parameters than other techniques that are as performant, and while not quite matching ensembles is oftentimes stronger than single-network uncertainty techniques.

**Weaknesses:**

I have a few concerns with this paper.

First:
The proposed model is extremely close to D'Angelo and Fortuin . Pairing this with shared backbones and spectral normalization is sound, but also not particularly impressive as an addition to the exploration space here.

Second:
The authors do not show enough ablations of the role of each part of their model.
How would they compare against combinations of their individual variations with baselines?
More importantly: what does spectral normalization really buy here?
How much is the backbone the key thing?

Third:
The authors evaluate a few different choices for their context dataset, but to my liking this is insufficient.
In their particle-based representation, parametric complexity is exchanged with evaluating context data, I will just call this a semi-parametric representation. As such, how much data is used for that semi-parametric representation and how that interacts with fidelity would be an important gradient to show here.
How stable is the model to varying that?
Which types of perturbations not heir images buy how much performance?
More importantly: the authors are proud of having reduced parameter count, but now require a battery of context data for each gradient step of their models. As such, evaluating how this interacts with memory requirements here is key.

In short: given that this paper jumps on the idea of representing BNNs functionally in a semi-parametric way, talking about the memory requirements for each type of computation and how that Pareto front varies seems as important as reporting the final parameter counts.

This might also help the authors justify their shared backbone more throughly: possibly the functional representation necessitates techniques to reduce memory footprint like shared backbones to more efficiently use GPU memory since the representation of f_backbone(context) is shared and as such does not incur memory overhead.

**Questions:**

I mentioned a lot of questions in the weaknesses tab.

Ablations of the pieces here.

A rigorous study of the context points and their properties and effects on this representation.

A study of the memory trade-offs when doing this.

In what scenarios is this type of inference beneficial compared to full ensembles?

I imagine there will be a regime of a certain dimensionality of networks or data complexity where the ensemble representation more efficiently captures uncertainty compared to the functional view.

I would enjoy seeing these talked about a lot more, they feel core to the paper's thesis.

Subject to seeing a significant change in the paper along these axes that offers more empirical learnings and insights for the reader given that the technical ideas are limited I would consider changing my score.

---

> ### Author Response · Authors · 2023-11-22
>
> Dear Reviewer,
>
>
> We appreciate your evaluation and thank you for your feedback. We acknowledge your observation related to the proximity of our proposed model to the work of D'Angelo and Fortuin, as we also mention in our paper.
> The main differences are the following: First, we show that the function space formulation is not restricted to deep ensembles and empirically analyze the benefits for improving a single network. Furthermore, we discuss the choice of context points for the repulsion loss evaluation and show that improvements are only achieved when using informative context points, e.g. CIFAR100 for CIFAR10. Evaluating the repulsion loss on batches of training data quickly leads to a degradation of in-domain (ID) accuracy. Although our ensemble head does not improve in-domain accuracy, the estimation of uncertainty for out-of-distribution samples is improved. Considering recent research questioning the effectiveness of deep ensembles ("Deep Ensembles Work, But Are They Necessary?"), we believe that looking at the problem from a function space point of view can be useful to further improve single networks with low computational effort in the forward pass.
>
> During training, we use the same batch size for the training data and the context points. By working with a single pre-trained base network, the size of the ensemble head is significantly reduced compared to ensembles of the base network. Loading context points in batches during ensemble head training results in minor overhead.
>
> Furthermore, we conducted ablation studies on the influence of the base network, spectral normalization, ensemble head architecture, and number of particles, reaffirming that the key factor remains the selection of context points.
>
> The additional ablation studies in the appendix are structured as follows:
> A4) Base network architecture
> A5) Ensemble head architecture
> A6) Number of particles
> A7) Choice of context points
>
> We hope that the additional experiments and discussions provide a more comprehensive understanding of our proposed model and its implications. Thank you for your time and consideration.

---

### Meta-Review · Area_Chair_F7Lz · 2023-12-06

**Metareview:**

This paper proposes to extend single-forward-pass uncertainty estimation methods in deep learning with multiple prediction heads, which are then trained using a particle-based VI approach. After an active discussion between authors and reviewers, all reviewers agreed to reject this paper. While the reviewers praised the importance of the problem and the strong empirical performance of the approach, they were critical of the novelty of the method, the ablation studies, and the selection of baselines. I believe that this could be a really interesting contribution and would encourage the authors to take the reviewer feedback seriously and resubmit a revised version of the paper in the future.

**Justification For Why Not Higher Score:**

see above

**Justification For Why Not Lower Score:**

N/A

---

### Decision · Program_Chairs · 2024-01-16

Reject